# Potential impact of carbonaceous aerosols on the Upper Troposphere and Lower Stratosphere (UTLS) during Asian summer monsoon in a global model simulation

Suvarna Fadnavis[1], Gayatry Kalita[1], K. Ravi Kumar[1], Blaz Gasparini[2] and Jui-Lin Frank Li[3]

[1]Indian Institute of Tropical Meteorology, Pune, India

[2]Institute for Atmospheric and Climate Science, ETH Zürich, Switzerland

[3]Jet Propulsion Laboratory, California Institute of Technology, Pasadena, California

**Abstract**

Recent satellite observations show efficient vertical transport of Asian pollutants from the surface to the upper level anticyclone by deep monsoon convection. In this paper, we examine the transport of carbonaceous aerosols including Black Carbon (BC) and Organic Carbon (OC) into the monsoon anticyclone using of ECHAM6-HAM, a global aerosol climate model. Further, we investigate impacts of enhanced (doubled) carbonaceous aerosols emissions on the UTLS from sensitivity simulations.

These model simulations show that boundary layer aerosols are transported into the monsoon anticyclone by the strong monsoon convection from the Bay of Bengal, southern slopes of the Himalayas and the South China Sea. Doubling of emissions of BC and OC aerosols, each, over the South East Asia (10°S - 50°N; 65°E - 155°E) shows that lofted aerosols produce significant warming in the mid/upper troposphere. These aerosols lead to an increase in temperature by 1K - 3 K in the mid/upper troposphere and in radiative heating rates by 0.005 K/day near the tropopause. They alter aerosol radiative forcing at the surface by -1.4 W/m$^2$; at the Top Of the Atmosphere (TOA) by +1.2 W/m$^2$ and in the atmosphere by 2.7 W/m$^2$ over

the Asian summer monsoon region (20ºN - 40°N, 60ºE - 120°E). Atmospheric warming increases vertical velocities and thereby cloud ice in the upper troposphere. An anomalous warming over the Tibetan Plateau (TP) facilitate the relative strengthening of the monsoon Hadley circulation and elicit enhancement in precipitation over India and north east China.

*Key words*: Aerosol radiative forcing; Black carbon and Organic carbon aerosols; ECHAM6-HAM; Upper Troposphere and Lower Stratosphere (UTLS); Asian Tropopause Aerosol Layer (ATAL).

## 1. Introduction

South East Asia (10°S–50°N; 65°E–155°E) being one of the most fast-growing population and economies which contributes significantly to the emission of global aerosol particles (Ramanathan and Crutzen, 2003; Lin et al., 2013). India and China are the two major contributors in South East Asia (Carmicael et al., 2009; Lin et al., 2014; Butt et al., 2016). Black Carbon (BC) and Organic Carbon (OC) are the important aerosol species as they contribute largely to the climate forcing (Penner et al., 1998; Chung and Seinfeld, 2002; Ramanathan and Carmichael, 2008; Hodnebrog et al., 2014), alter the energy balance in the atmosphere and the global water cycle (Solomon et al., 2007). Recent studies show that their impacts on local meteorology and monsoon circulation are significantly large (Ackerman et al., 2000; Ramanathan et al., 2001a, 2001b; Lelieveld et al., 2001; Menon et al., 2002; Manoj et al., 2011). BC and OC together account for more than 60 % of the AOD (Chin et al., 2009; Streets et al., 2009).

There is ever growing concern for rapidly increasing anthropogenic emissions of carbonaceous aerosols namely BC and OC. Global emissions of BC have almost doubled during the past century (Baron et al., 2009). Developing countries in Asia, e.g. India and China produce BC emissions at high growth rates. These countries together produced about 40% of total world BC emissions from combustion (Kopp and Mauzerall, 2010). The estimated growth of BC is 46 % (33% in OC) over China and 41% (35% in OC) over India during 2000 to 2010 (Lu et al., 2011). On a regional scale, their emissions are high over densely populated Indo-Gangetic Plains in India and eastern China (Kumar et al., 2011; Lelieveld, 2001; Gautam et al., 2011; Fadnavis et al., 2013; Zhang et al., 2015) (see Fig. 1).

Majority of BC and OC aerosols are formed by incomplete combustion (Satheesh and Ramanathan; 2000; Carmicael et al., 2009). The important emission sources of BC aerosols are diesel vehicles, exhaust from coal-based power plants, exhaust from industries, forest fires and residential bio-fuel and fossil-fuel combustion. The OC aerosols are produced from fossil fuel and biofuel burning and natural biogenic emissions. Biogenic carbonaceous aerosol consist of plant debris, pollen, fungal spores, and bacteria (Jacobson et al., 2000; Bond et al., 2004) and secondary organic aerosol from oxidation of volatile organic compounds (VOCs) (Solomon et al., 2007).

Recent satellites, Cloud Aerosol Lidar and Infrared Path finder Satellite Observation (CALIPSO) (Vernier et al., 2011; Thomason and Vernier, 2013), Stratospheric Aerosol and Gas Experiment II (SAGE) (Thomason and Vernier, 2013) and Balloonsonde (Vernier, et al., 2015) observations show Asian Tropopause Aerosol Layer (ATAL) near the tropopause persisting during the monsoon season (June-September). Satellite observations reveal transport of trace gasses (CO, PAN, $H_2O$ HCN) into the upper level monsoon anticyclone by deep monsoon convection (Park et al., 2009; Randel et al., 2010, Kunz et al.,2010; Ploeger et al., 2011; 2012; 2013; Fadnavis et al., 2014; 2015, Govardhan et al., 2017). Moreover, both back trajectory analysis based on CALIOP observations (Vernier et al., 2015) and modeling studies (Fadnavis et al., 2013) indicate that deep monsoon convection transports boundary layer aerosols into the UTLS. A Civil Aircraft for Regular Investigation of atmosphere Based on an Instrument Container (CARBIC) measurements show aerosols at the lower levels in the ATAL contain higher levels of carbonaceous and sulfate aerosols. The ratio of carbon to sulfur is ~4.0 with concentrations of carbon ~36 $ng/m^3$ and sulfur ~ 13 $ng/m^3$ in the Asian upper troposphere during August 2006, 2007 and 2008 (Vernier et al., 2015). Carbonaceous aerosols in the upper

troposphere lead to atmospheric heating due to their absorptive properties which may subsequently alter the atmospheric thermal structure and cloud amounts. Higher concentrations of carbonaceous aerosols in the ATAL may significantly alter thermal structure of the UTLS and therefore the underlying monsoon circulation (Meehl et al., 2008; Kloster et al., 2009). The ATAL may affect the radiative forcing regionally. Vernier et al., (2015) reported that the ATAL had exerted a short-term regional forcing at the top of the atmosphere ~ -0.1 W/$m^2$ during past two decades.

Asian Summer Monsoon (ASM) has a major impact on agriculture, water resources, and economy and social life. Therefore it is important to study the impact of fast-growing Asian emission of carbonaceous aerosols on monsoon precipitation. However, there are a few studies reporting the impacts of carbonaceous aerosols on precipitation over India (Meehl et al., 2008; Wang et al., 2009; Ganguly et al., 2012) and China (Guo et al., 2013; 2015). Since convective transport (during the monsoon season) inter-links tropospheric processes with the UTLS (Randel et al., 2010; Vogel et al., 2011, 2015; Fadnavis et al., 2013), it is essential to understand impacts of boundary layer emissions on the UTLS. To our knowledge, transport of carbonaceous aerosols from the boundary layer to upper troposphere, their impacts on the UTLS and connecting monsoon circulation are not explored in detail. In this study, we address the question of the impact of rapidly growing emissions of carbonaceous aerosols (BC and OC) on the thermal structure of the UTLS, monsoon transport processes and rainfall over India and China. We perform control and sensitivity simulations using the ECHAM6-HAM aerosol climate model. In sensitivity experiment, we doubled anthropogenic emissions of BC and OC, each, over the South East Asia (10°S-50°N; 65°E-155°E). The paper is organized as follows; in Section 2 model simulations and satellite observations are described. The transport processes are discussed in

Section 3. The impact of enhanced carbonaceous aerosols emissions on the UTLS and monsoon precipitation are described in Section 4, followed by conclusions given in Section 5.

## 2. Model simulations and satellite data analysis

### 2.1 Experimental setup and model simulations

The fully coupled aerosol-climate model ECHAM6-HAM (version echam6.1.0-ham2.1-moz0.8) used in this study comprises the general circulation model ECHAM6 (Stevens et al., 2013) coupled to the aerosol sub-module Hamburg Aerosol Model (HAM) (Stier et al., 2005, Zhang et al., 2012). HAM predicts the evolution of sulfate (SU), black carbon (BC), particulate organic matter (POM), sea salt (SS), and mineral dust (DU) aerosols. The size distribution of aerosol population being described by seven log-normal modes with prescribed variance as in the M7 aerosol module (Vignati et al., 2004; Stier et al., 2005; Zhang et al., 2012). Moreover, HAM uses the two-moment cloud microphysics scheme in which the nucleation scavenging of aerosol particles by acting as cloud condensation nuclei or ice nuclei, freezing and evaporation of cloud droplets and melting and sublimation of ice crystals is treated explicitly (Lohmann et al., 2007, Lohmann et al., 2010, Neubauer et al., 2014). The anthropogenic and fire emissions of $SO_2$, BC, and OC are based on the AEROCOM-ACCMIP-II emission inventory. The anthropogenic emissions are based on Lamarque et al., (2010). The biomass burning emissions are from GICC 1850 - 1950 (Mieville et al., 2010), RETRO 1960-1990 (Schultz et al., 2008) and GFED v2 (1997 - 2008) (van der Werf et al., 2006). Biogenic emissions are derived from MEGAN (Guenther et al., 1995) and fossil fuel sources are provided by the ACCMIP inventory (Lamarque et al., 2010). In the model, biogenic OC is directly inserted via emissions. Secondary organic aerosol (SOA) emissions are as described by Dentener et al. (2006). The emissions of

sea salt (Guelle et al., 2001 and Stier et al., 2005) and dust (Tegen et al., 2002; Cheng et al., 2008) are computed interactively.

The model simulations are performed at the spectral resolution of T63. This spectral representation is associated with a horizontal resolution of 1.875° x 1.875° on a Gaussian grid and a vertical resolution of 47 levels spanning from the surface up to 0.01 hPa. The simulations have been carried out at a time step of 20 minutes. AMIP sea surface temperature (SST) and sea ice cover (SIC) are used as lower boundary conditions. Note that our base year for aerosol and trace gas emissions is 2000. Each simulation was performed for the 30 years from January 1979 to December 2009. We analyze simulated data for 20 years (1989-2009) considering initial ten years as spin-up time. Emissions are the same in each simulation, and meteorology varied because of different monthly sea surface temperature (SST) and sea ice (SIC) data. Most of the models underestimate BC and OC mass concentrations observed over Asia (Mao et al., 2011; Bond et al., 2013; Butt et al., 2016; Winiger et al., 2016). Bond et al. (2013) have suggested that global atmospheric absorption attributable to black carbon is too low in many models and should be increased by a factor of three. Butt et al. (2016) obtained better predictions when residential carbonaceous emissions were doubled. We performed control experiment (CTRL) in which we kept our emissions at the baseline levels (the year 2000) and a sensitivity experiment (DEMISS) in which we doubled emissions of BC and OC, over the South East Asian region (65°E - 155°E; 10°S - 50°N). We compare CTRL simulation with DEMISS and analyze the impacts of doubled carbonaceous emissions (BC and OC together) on the UTLS and rainfall during ASM season (June - September).

### 2.2 Satellite measurements

#### 2.2.1 The Tropical Rainfall Measuring Mission (TRMM)

The Tropical Rainfall Measuring Mission (TRMM) is a joint National Aeronautics and Space Administration (NASA) - Japan Aerospace Exploration (JAXA) satellite mission to monitor the tropical and subtropical precipitation and estimate its associated latent heat. TRMM was launched in 1997 from Tanegashima space center in Japan. The rainfall measuring instruments on the TRMM satellite include an electronically scanning radar Precipitation Radar (PR), (operating at 13.6 GHz), TRMM microwave image (TMI), a 9 channel passive microwave radiometer (which records radiation at the 10.65, 19.35, 37.0, 85.5 (V and H) and 21.3 (V) GHz), and Visible and Infrared Scanner (VIRS) with five operating channels (Kummerow et al., 1998). The 3B42 algorithm produces TRMM adjusted merged infrared precipitation rate and root mean square (RMS) precipitation error estimates (Huffmen et al., 2007). The algorithm combines multiple independent precipitation estimates from the TMI, Advanced Microwave Scanning Radiometer for Earth Observing Systems (AMSR-E), Special Sensor Microwave Imager (SSMI), Special Sensor Microwave Imager/Sounder (SSMIS), Advanced Microwave Sounding Unit (AMSU), Microwave Humidity Sounder (MHS), and microwave-adjusted merged geo-infrared (IR). The final 3B42 precipitation (in mm $hr^{-1}$) estimates have a 3-hourly temporal resolution and a 0.25-degree by 0.25-degree spatial resolution. TRMM precipitation can be obtained from https://disc2.gesdisc.eosdis.nasa.gov/data/TRMM_L3/TRMM_3B42.7/. 3-hourly precipitation data are averaged to obtain daily mean. Then, seasonal mean (June-September) is computed from daily mean data. Further, seasonal mean data is averaged for 20 years (1997-2016) to obtain climatology of the monsoon season.

### 2.2.2 CloudSat and Cloud-Aerosol Lidar Infrared Pathfinder Satellite Observations (CALIPSO)

Cloud–Aerosol Lidar and Infrared Pathfinder Satellite Observation (CALIPSO) and CloudSat are two A-Train constellation satellites, launched together in April 2006. They provide information related to the role of cloud and aerosol in the Earth's climate system and radiative imbalance of the atmosphere. The Cloud Profiling Radar (CPR) on board of CloudSat satellite is a 94-GHz nadir-looking radar which measures the power backscattered by clouds as a function of distance. It provides information on cloud abundance, distribution, structure, and radiative properties (Stephens et al., 2008). The Cloud-Aerosol Lidar with Orthogonal Polarization (CALIOP) is an elastically backscattered active polarization sensitive Lidar instrument onboard CALIPSO. The CALIOP transmit laser light simultaneously at 532 nm and 1064 nm at a pulse repetition rate 20.16Hz. The Lidar receiver subsystem measures backscatter intensity at 1064 nm and two orthogonally polarized components of 532 nm backscatter signal that provides the information on the vertical distribution of aerosols and clouds, cloud particle phase, and classification of aerosol size (Winker et al., 2010; Powel et. al., 2013). In this study, we use Ice water content (IWC) dataset from the combined measurement of CloudSat and Calipso (2C-ICE_L3_V01) for the period 2007-2010. The 2C-ICE cloud product is an ice cloud retrieval derived from the combination of the CloudSat radar and CALIPSO Lidar, using a variational method for retrieving profiles of the IWC in ice clouds (Deng et al., 2013). The details of the data retrieval method are explained in Li et al., (2012). IWC data has been averaged for the monsoon season and period 2007-2010 to obtained seasonal climatology.

### 2.3 Comparison with measurements

We compare CTRL simulated BC concentrations with in-situ measurements reported by Babu et al., (2011) over Hyderabad (17°.48' N; 78°.40'E) in India on 17 March 2010 during pre-monsoon season. Babu et al., (2011) obtained BC measurements using aethalometer installed in the hydrogen-inflated balloon. For comparison, monthly mean simulated BC concentrations for March are extracted at the grid centered at 17°N, 78°E. Figure 2a shows the comparative analysis of model simulated BC and in situ measurements of BC. It can be seen that peaks near 4 km and 8.5 km are not reproduced by the model simulations. Balloon borne measurements show high values of BC concentrations ~12 μg $m^{-3}$ near 4-5 km altitude whereas the model simulations show values of ~0.4-1μ $gm^{-3}$. These peaks and fluctuation in BC profile indicate an influence of meteorology on that day. The model could not reproduce such peaks as simulations were not forced by the meteorology; while we show a monthly mean profile (model output is written at every month). It must be mentioned that the vertical profile of simulated BC is over a wider spatial grid (1.8° x 1.8°) whereas balloonsonde measurements by Babu et. al., (2011) are at a single station. The model underestimates BC concentrations by ~2.1 μg $m^{-3}$ near 2 km - 4 km and ~0.8 μg $m^{-3}$ near 6 km -7.5 km. Tripathi et al. (2007) reported BC concentrations ~8 μg $m^{-3}$ - 4 μg $m^{-3}$ between the surface to 2 km at Kanpur (80°.20'E, 26°.26'N). Simulated BC concentrations at Kanpur show similar values (7.5 μg $m^{-3}$ - 3 μg $m^{-3}$).

Figures 2b and 2c show the vertical distribution of cloud ice obtained from CTRL simulation and climatology of seasonal mean from combined measurement of CloudSat and CALIPSO (2C-ICE) (2007-2010) respectively, averaged for the monsoon season (June-September) and ASM region (60°E - 110° E;15°N - 40°N). It can be seen that simulated (3 mg/kg -10 mg/kg) and observed cloud ice (5 mg/kg - 17 mg/kg), both, show high amounts in the upper troposphere (450

hpa - 200 hPa) over the ASM region. The model simulations show maximum (7 mg/kg - 10 mg/kg) at ~350 hPa - 250 hPa over 80ºE - 100ºE while satellite observations (12 mg/kg – 17 mg/kg) show it at ~450 hPa-200 hPa over ~80ºE - 120ºE. These differences may be related to uncertainties in satellite observations (Deng et al., 2010) and model biases, e.g., the model does not consider large ice particles unlike the cloud ice measurement from CloudSat and CALIPSO. The total ice water mass estimate from 2C-ICE, combine measurements from CALIPSO Lidar depolarization which is sensitive to small ice particle (i.e., cloud ice represented in GCMs) while CloudSat radar which is very sensitive to larger ice particles (i.e., precipitating ice or snow). In most global climate models including all the CMIP3 and most of the CMIP5, only small particles (i.e., cloud ice) are represented prognostically. The mass of large ice particles (about two-third of total ice) and their radiative effects, however, are not included (e.g., Li et al., 2012; 2013).

We compare simulated (CTRL) seasonal mean (June-September) precipitation with TRMM precipitation climatology (1997-2016). Figures 2d and 2e show the distribution of precipitation as obtained from CTRL simulation and TRMM respectively. It can be seen that general spatial pattern of precipitation simulated by the model is in good agreement with the TRMM. The model could reproduce high amounts of precipitation over the Bay of Bengal, the South China Sea, and the Western Ghats, in agreement with a numbers of past studies (Kang et al., 2002; Wang and Linho, 2002; Zveryaev and Aleksandrova, 2004; Hirose and Nakamura, 2005; Xie et a.l., 2007; Takahashi, 2016). However, model underestimates the rainfall over northern India and the Western coast of India by ~ 2 mm/day - 10 mm/day and overestimates over the Tibetan Plateau (TP) and the South China Sea by ~5 mm/day - 12 mm/day. It may be related uncertainties in emissions, transport errors, and model coarse resolution.

## 3. Transportation of aerosol to the UTLS

Figure 3a depicts the vertical distribution of carbonaceous aerosols averaged over North India (75°E - 100°E; 25°N - 45°N) during the annual cycle as obtained from CTRL simulation. It shows elevated levels of aerosols (BC and OC together) from the surface to the tropopause during pre-monsoon (March-May) and monsoon seasons. It shows a layer of carbonaceous aerosols (~5 ng/m$^3$) in the upper troposphere ~170hPa - 100hPa. A layer of aerosols in the upper troposphere is also observed by satellite (SAGE II, CALIPSO) and ground-based Lidar measurements during the monsoon season (Vernier et al., 2011; Thomason and Vernier, 2013; Fadnavis et al. 2013; He et al., 2014). Over the TP this aerosol layer extends above the tropopause (18-19 km) (He et al., 2014).

A prominent feature in the UTLS over the ASM region during the summer season is a large anticyclone. Satellite observations show a persistent maximum in trace gases (CO, $H_2O$, PAN, HCN, $CH_4$, etc) (Li et al., 2005a; Randel and Park 2006, Fu et al., 2006; Park et al., 2008; 2009, Randel et al., 2010; Fadnavis et al., 2013, 2014, 2015) and aerosols (Tobo et. al., 2007; Vernier et. al., 2011; Thomason and Vernier, 2013; Yu et al., 2015) in the ASM anticyclone. Figure 3b exhibits the distribution of seasonal (June-September) mean carbonaceous aerosols (BC and OC together) from CTRL simulation in the anticyclone (~100 hPa). In agreement with previous studies (Tobo et. al., 2007 Vernier et. al., 2011), Fig. 3b also shows confinement of high carbonaceous aerosols concentration (>5 ng/m$^3$) within the anticyclone. The wind vector at 100hPa depicts the extent of the anticyclone (20°E-120°E and 15°N-40°N).

Previous studies from model simulations and trajectory analysis show that rapid transport of trace gases and aerosols from Asian boundary layer into the anticyclone is closely linked with the deep ASM convection (Li et al, 2005; Randel and Park, 2006; Park et al., 2007; Park et al,

2009; Xiong et al., 2009; Fadnavis et al, 2013, 2014 , 2015). We plot longitude-pressure and latitude-pressure cross sections of carbonaceous aerosol from CTRL simulations to understand their transport. Figure 3c displays seasonal mean longitude-pressure variation of carbonaceous averaged over 15°N-35°N, along with wind vectors. It indicates that they are lifted up from the Bay of Bengal, Indo-Gangetic Plains (70ºE-90ºE) and South China Sea (110ºE-130ºE) into the anticyclone increasing the aerosol concentration to 4-6 ng/m$^3$ in the UTLS (above 200hPa) across 40°E-110°E. Transport from southern slopes of Himalaya is evident in Figs. 3d. Figures 3e and 3f show the condensed cloud water (both liquid and ice). Its maxima point out areas of frequent deep convective activity over the Bay of Bengal and the South China Sea (Fig. 3e) and the southern flanks of the Himalayas (Fig. 3f). Thus transport of carbonaceous aerosols (seen in Figs. 3c and 3d) from these regions into the upper level anticyclone may be due to deep monsoon convection. Pollution transport (CO, HCN, NOX, PAN) from the Asian region to the UTLS due to monsoon convection is also reported by Park et al. (2007), Randel et al. (2010), Fadnavis et al., (2014, 2015). Figures 3c and 3d show that a fraction of aerosols crosses the tropopause and enters into the lower stratosphere. It may be due to large scale upward motion within the anticyclone shown by the wind vectors. Recently, trajectory analysis showed that air masses within the anticyclone are transported into the lower stratosphere in the northern subtropics (Garny and Randel, 2016).

We analyze the vertical profile of anomalies of carbonaceous aerosols obtained from a difference between DEMISS and CTRL simulations. Longitude-pressure and latitude-pressure cross sections of the anomalies are shown in Figs. 4a and 4b respectively. Enhanced anomalies are seen along the transport pathways, e.g., from the Bay of Bengal, the South China Sea and southern flanks of the Himalayas into the anticyclone. They show an enhancement of nearly 4

ng/m$^3$ relative mass of aerosol near the tropopause and part of it (>2 ng/m$^3$) enters the lower stratosphere.

## 4. Impact of enhanced carbonaceous aerosols emissions

### 4.1 Impact on radiative forcing and heating rates

BC and OC aerosols absorb and scatter radiation, resulting in heating of the atmosphere and reduction of solar radiation reaching the Earth's surface (Penner et al., 1998). The global mean estimated cumulative (since 1970) BC radiative effect is +0.3 W/m$^2$ while OC emitted from fossil fuels is estimated to be -0.1 W/m$^2$ (Myhre et al., 2013). The presence of BC aerosols can change the sign of forcing from negative to positive (Haywood and Shine, 1997).

The convectively transported carbonaceous aerosols may alter radiative forcing, heating rates, temperature, and vertical velocities in the UTLS. The carbonaceous aerosol can affect the radiative energy balance of the atmosphere directly by scattering and absorbing solar radiation and indirectly by acting as cloud condensation nuclei (Rosenfield 2000). This indirect forcing is neglected in our model simulations as these aerosols are not considered to act as cloud condensation nuclei. Anomalies in aerosol forcing estimated from DEMISS simulation against CTRL (i.e., DEMISS - CTRL) are averaged for the monsoon season and ASM region (see Table-1). Seasonal mean anomaly of aerosol forcing is +1.2 W/m$^2$ at the top of the atmosphere (TOA) and -1.4 W/m$^2$ at the surface. The atmospheric radiative forcing is computed from the difference between forcing at TOA and surface. The resultant anomaly of atmospheric aerosol radiative forcing is ~2.6 W/m$^2$. It represents the energy trapped in the atmosphere due to the presence of higher amounts of carbonaceous aerosols. Babu et al., (2002) reported BC radiative forcing +5 W/m$^2$ at the TOA and at the surface -23 W/m$^2$ in Bangalore (13ºN, 77ºE), India. Badarinath and

Latha, (2006) obtained BC radiative forcing of +9 W/m$^2$ at the TOA and -33 W/m$^2$ at the surface at Hyderabad (78ºE, 17ºN), India.

The resulting shortwave plus longwave atmospheric forcing due to doubled carbonaceous aerosol will translate to a significant atmospheric heating (Babu et al., 2002). We obtained anomalies in total heating rates (HR) due to carbonaceous aerosols (DEMISS - CTRL). Figures 4c and 4d show longitude-pressure (averaged for 15ºN-35ºN) and latitude-pressure (averaged for 80ºE-110ºE), cross sections of HR anomalies during the monsoon season (wind anomalies are plotted over HR anomalies). Enhanced carbonaceous aerosols emissions increase HR near the surface. High emissions from Indo-Gangetic Plains (70ºE - 90ºE, 25ºN - 35ºN) cause anomalous heating (0.05 K/day) in the lower troposphere (1000 hPa - 600 hPa). Positive anomalies of HR can be seen along the pathway through which carbonaceous aerosols are transported into the anticyclone. It can be seen that carbonaceous aerosols have increased HR by ~0.003 K/day - 0.005 K/day at tropopause level in the AMS region in comparison with CTRL simulations (0.006 K/day - 0.01 K/day). Park et al. (2007) estimated net HR rates near the tropopause (averaged over 60°E-150ºE) ~0.2 K/day - 0.6 K/day during the monsoon season. In comparison, HR estimated from CTRL simulation ~ are less (0.1 K/day – 0.25 K/day) over the same region. Radiative heating of the tropopause region increases the vertical motion and transport into the lower stratosphere (Gettleman et al., 2004). Carbonaceous aerosols enhancement (> 2 ng/m$^3$) in the lower stratosphere seen in Figs. 4a and 4b may be due to increase in vertical motion in response to enhanced aerosol HR. This indicates that aerosols induce positive feedback in vertical transport.

### 4.2 Impact on temperature and precipitation

Further, we analyze changes in temperature induced by doubled carbonaceous aerosol emissions. Figures 4e and 4f show the longitude-pressure (averaged over 15ºN - 35ºN) and latitude-pressure (averaged over 60°E - 110°E) cross sections of temperature anomalies. These aerosols induce significant warming in the mid-troposphere (500 hPa -300 hPa) over the ASM region and a striking warm core like feature of anomalous warming (~3K) in the mid-upper troposphere over the TP (70ºE - 90ºE, 30ºN - 45°N) (Fig. 4f). The warm core over the TP plays an important role in enhancing the ASM circulation (Flohn 1957; Yanai et al., 1992; Meehl, 1994; Li and Yanai, 1996; Wu and Zhang, 1998) (discussed later in this section). Figure 4e shows cooling near the tropopause in the anticyclone with a small patch of positive anomalies over the TP (80º-100°E). During the monsoon season, cold temperatures in the UTLS overlie warm mid-troposphere (Randel and Park 2006; Park et al., 2007). Our model simulations show that doubling of carbonaceous aerosol emissions amplifies the mid-tropospheric warming and cooling near the tropopause.

During Northern hemispheric summer, heating over the TP maintains a large-scale thermally driven vertical circulation (Yanai et al., 1992). The analysis of simulated vertical velocities shows that carbonaceous aerosols induce positive anomalies over the southern TP and Indo-Gangetic plains (Figs 5a and 5b). Thus carbonaceous aerosols amplify warming (Fig. 4e and Fig. 4f) and enhance ascending motion over these regions. Previous studies (Rajagopalan and Molnar, 2013, Vinoj et al., 2014) have reported that the warm ascending air above the TP gradually spreads southward and descends over the northern Indian Ocean. The south-westerly winds at the surface, on the other hand, complete the monsoon Hadley cell. This local circulation system releases latent heat and further maintains the Tibetan warm core. Thus heating over the

TP leads to increased Indian summer monsoon rainfall by enhancing the cross-equatorial circulation and concurrently strengthening both the Somali Jet and the westerly winds that bring rainfall to India. Goswami et al., (1999) also reported that there is a strong correlation between monsoon Hadley circulation and precipitation. Figure 5c shows that carbonaceous aerosols strengthen the monsoon Hadley circulation, ascending motion over 10ºN - 20ºN and descending over 0º-10ºS.

Thus Figs. 4a-4f and Figs 5a-5c suggest that enhanced emissions of carbonaceous aerosols increase the HR, and amplify warm anomalies in the middle troposphere and cold anomalies near the tropopause. Aerosol induced warming elicits enhancement in vertical velocities. These aerosols induce an anomalous warming over the TP which in turn strengthens the monsoon Hadley circulation. Previous studies (Meehl et al., 1994; Krishnamurthy and Achuthavarier 2002) have explained the mechanism of strengthening of the monsoon Hadley circulation facilitate enhanced precipitation over the Indian region. Consequently, aerosol (carbonaceous) induced precipitation anomalies are positive over the Indian region (1 mm/day - 4 mm/day) (Fig. 5d). Strong positive anomalies (2 mm/day - 3.5 mm/day) are located over North India, the Bay of Bengal, Western coast of India and foothills of Himalaya. There is an enhancement in precipitation over North east China (0.2 mm/day - 2 mm/day) and some parts of central and south China (0.2 mm/day - 1 mm/day). In agreement with the present study, aerosol-climate modeling studies by Wang et al., (2004, 2007) also show enhancement in Indian summer monsoon precipitation due to black carbon direct radiative forcing. Increases the Indian summer monsoon precipitation due to the loading of absorbing aerosol (BC and dust) has been reported in the past (Lau and Kim., 2006; Vinoj et al., 2014; Fadnavis et al., 2016). However, a mix response is portrayed by Ganguly et al. (2012). Their ocean-atmosphere coupled model show

reduction in precipitation over the western coastline of the Indian peninsula and increase over north western part of Indian subcontinent. Reduction in precipitation is attributed to anthropogenic local and remote aerosols. These differences may be due to different model-set up, present study gives impact of doubled Asian carbonaceous aerosol emissions using Aerosol-atmosphere-climate model. While, Ganguly et al. (2012) reports response of all anthropogenic and biomass burning aerosols using a coupled atmosphere-slab ocean model simulations.

### 4.3 Impact on water vapor, cloud ice

Recently from satellite observations, Park et al., (2007) have shown that water vapor in the upper troposphere (~216 hPa) varies coherently with deep monsoon convection both temporally and spatially. Transport of high water vapor in the UTLS by the monsoon convection has been reported in the past (Randel et al., 2001; Gettelman et al., 2004; Dessler and Sherwood, 2004; Fu et al., 2006; Randel and Park, 2006, Braesicke et al., 2011; Ploeger et al., 2013). We analyze the difference in water vapor anomalies (DEMISS - CTRL) to understand the impact of doubled Asian carbonaceous aerosol emissions on the transport of water vapor in the UTLS. Figures 6a and 6b show an increase in water vapor transport in the upper troposphere and lower stratosphere (0.1 ppmv - 2 ppmv). Water vapor anomalies ~8 ppmv - 20 ppmv are seen near 200 hPa and ~0.1 ppmv - 0.8 ppmv near the tropopause. Fadnavis et al. (2013) reported an increase in water vapor (~ 0.1 ppmv - 10 ppmv) in the UTLS in response to increasing in aerosols which are in agreement with the current study. In the past, Gettleman et al. (2004), Fu et al. (2006), Fadnavis et al., (2013), Garny and Randel (2016) also reported transport of water vapor above the tropopause into the lower stratosphere during the monsoon season. Enhanced aerosol emissions increase water vapor transport into the lower stratosphere by enhancing heating rates, mid/upper tropospheric warming, and vertical velocities.

In addition to thermal and dynamical impact, aerosols in the UTLS also largely influence the formation and microphysical properties of cirrus clouds. Cirrus clouds have a great impact on radiation and intensity of the large-scale tropical circulation (Randall et al., 1989; Ramaswamy and Ramanathan, 1989; Liu et al., 2003). Figures 6c - 6f show longitude-pressure and latitude-pressure cross sections of anomalies of cloud ice and Ice Crystal Number Concentration (ICNC). These figures show enhancement of anomalies of cloud ice (by 0.4 $mg/m^3$ - 1 $mg/m^3$) and ICNC (by 0.08 1/mg) occurrence in the upper troposphere (350 hPa - 100 hPa). Maximum increase (cloud ice by 0.6 $mg/m^3$ and ICNC by 0.08 $m^{-3}$) is seen in the 20ºN - 30ºN where stronger upwelling motion prevails (Figs. 6d and 6f). A fraction of positive anomalies of ICNC are seen near the tropopause indicating entrainment into the lower stratosphere. Positive anomalies in cloud ice and ICNC (in the upper troposphere) may be due to enhancement in ASM deep convection (increase in heating rates, mid/upper tropospheric temperature, vertical velocity, and monsoon Hadley circulation) induced by the doubling of carbonaceous aerosols emissions.

## 5. Conclusions

In this paper, we investigated impacts of enhanced Asian (65°E - 155°E; 10°S - 50°N) carbonaceous aerosols on the UTLS, underlying monsoon circulation and precipitation over India and China using a state of the art aerosol-climate model. We performed sensitivity experiments for doubling of carbonaceous aerosol over the Asian region.

To validate the model simulations, we compare simulated BC vertical profile with observations from aethalometer launched on Balloonsonde at Hyderabad (78°E, 17ºN) on 17 March 2010 in pre-monsoon season; seasonal mean of simulated cloud ice content with climatology of combined measurements from CloudSat and CALIPSO (2007-2010); and simulated precipitation with climatology of TRMM observations (1997-2016). Comparison of the simulated vertical

profile of BC aerosols with the balloon borne aethalometer measurements at Hyderabad (17 March 2010), shows that the model underestimates BC concentrations by ~2.1 μg $m^{-3}$ ~0.8 μg $m^{-3}$ in the troposphere (4-8 km) during the pre-monsoon season. The spatial patterns of the simulated season mean (June - September) precipitation are comparable with climatology of TRMM precipitation (1997-2016) and cloud ice with combined measurements from CloudSat and CALIOP (2007-2010) respectively. Simulated cloud ice is underestimated 2 mg/kg - 7 mg/kg in the UTLS (60°E - 120°E; 15ºN - 40°N) during the summer monsoon season.

Our model simulations show that monsoon convection over the Bay of Bengal, the South China Sea and Southern flanks of the Himalayas transport Asian carbonaceous aerosol into the UTLS. A persistent maximum of carbonaceous aerosols is seen within the anticyclone throughout the ASM season, and a fraction of these aerosols enter the lower stratosphere. Doubling emissions of carbonaceous aerosol over the Asian region leads to their enhancement (by 4-6 ng/$m^3$) in the UTLS. They alter aerosol radiative forcing at the surface by -1.4 W/$m^2$; at the TOA by +1.2 W/$m^2$ and in the atmosphere by 2.7 W/$m^2$. Positive anomalies of heating rates are seen along the pathway through which aerosols are transported into the anticyclone. These carbonaceous aerosols increase heating rates in the anticyclone (~100 hPa) by 0.003 K/day to 0.005 K/day. They induce significant warming (temperature increases by 1-3K) in mid/upper troposphere over the ASM region. An anomalous in-atmospheric warming enhances vertical velocities and thereby cloud ice (by 0.4-1 mg/$m^3$), ICNC (by 0.08 1/mg). A significant increase in water vapor transport in the upper troposphere (0.5-10 ppmv) and lower stratosphere (0.1 ppmv - 0.5 ppmv) is apparently related to the mid/upper tropospheric warming. Doubling of carbonaceous aerosols emissions enhance warming over the TP (~3K) and amplify cold anomalies near the tropopause (-0.1K - -1K). An anomalous warming over the TP enhances the

monsoon Hadley circulation and elicits an enhancement in precipitation over India (1-4 mm/day) and eastern China (0.2 mm/day - 2 mm/day). In agreement with the present study, aerosol-climate modeling studies by Wang et al., (2004, 2007) also show enhancement in Indian summer monsoon precipitation due to black carbon direct radiative forcing. Observational evidences also show that heavy loading of absorbing aerosols (BC and Dust) over the Indian subcontinent facilitate enhancement of monsoon rainfall over India (Lau and Kim, 2006; Vinoj et al., 2014). However, a mixed response, a regional increase (North western India) /decrease (Indian Peninsula and eastern Nepal) in precipitation in response to anthropogenic and biomass burning aerosol emissions is reported by Ganguly et al., (2012). These results differ from the present study. It may be due to different model-set up, present study gives impact of doubled Asian carbonaceous aerosol emissions using Aerosol-atmosphere-climate model. While, Ganguly et al. (2012) reports response of all anthropogenic and biomass burning aerosols using a coupled atmosphere-slab ocean model simulations.

We note that a realistic future emission scenario includes also increasing emissions of sulfate aerosols and the response of climate and circulation to increasing $CO_2$ concentrations, which might interplay with the presented results and lead to different dynamical and climatic responses. Moreover, in future, we propose to re-evaluate the studies by using an aerosol model coupled to the interactive chemistry, microphysics, the regional model with a better resolution of the complex orography over Himalayas/TP, etc. Notwithstanding this, the work provides valuable insight into the influence of growing Asian carbonaceous aerosols emissions on the UTLS, connecting monsoon processes and precipitation in the Asian summer monsoon region.

*Acknowledgement*: Authors acknowledges with gratitude Dr. Krishnan, Executive Director of CCCR, IITM, for his encouragement during the course of this study and the High Power Computing Centre (HPC) in IITM, Pune, India, for providing computer resources.

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

Table-1: ECHAM6 HAM simulated total (shortwave and long wave together) radiative forcing ($W/m^2$) and averaged over ASM region

| Model Run | TOA | Surface | Atmosphere |
|---|---|---|---|
| DEMISS | -4.2 | -12.1 | 7.9 |
| CTRL | -5.4 | -10.6 | 5.2 |
| Anomalies | 1.2 | -1.4 | 2.7 |

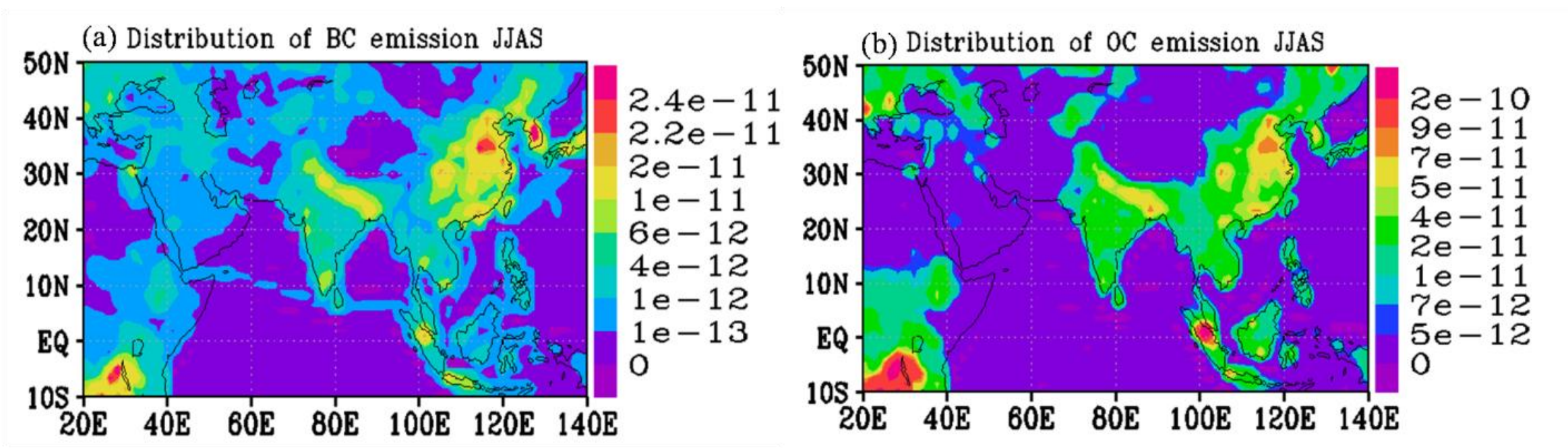

Figure1: Distribution of emission mass flux (kg $m^{-2}$ $s^{-1}$) averaged for the monsoon season (June-September) for **(a)** BC and **(b)** OC aerosols.

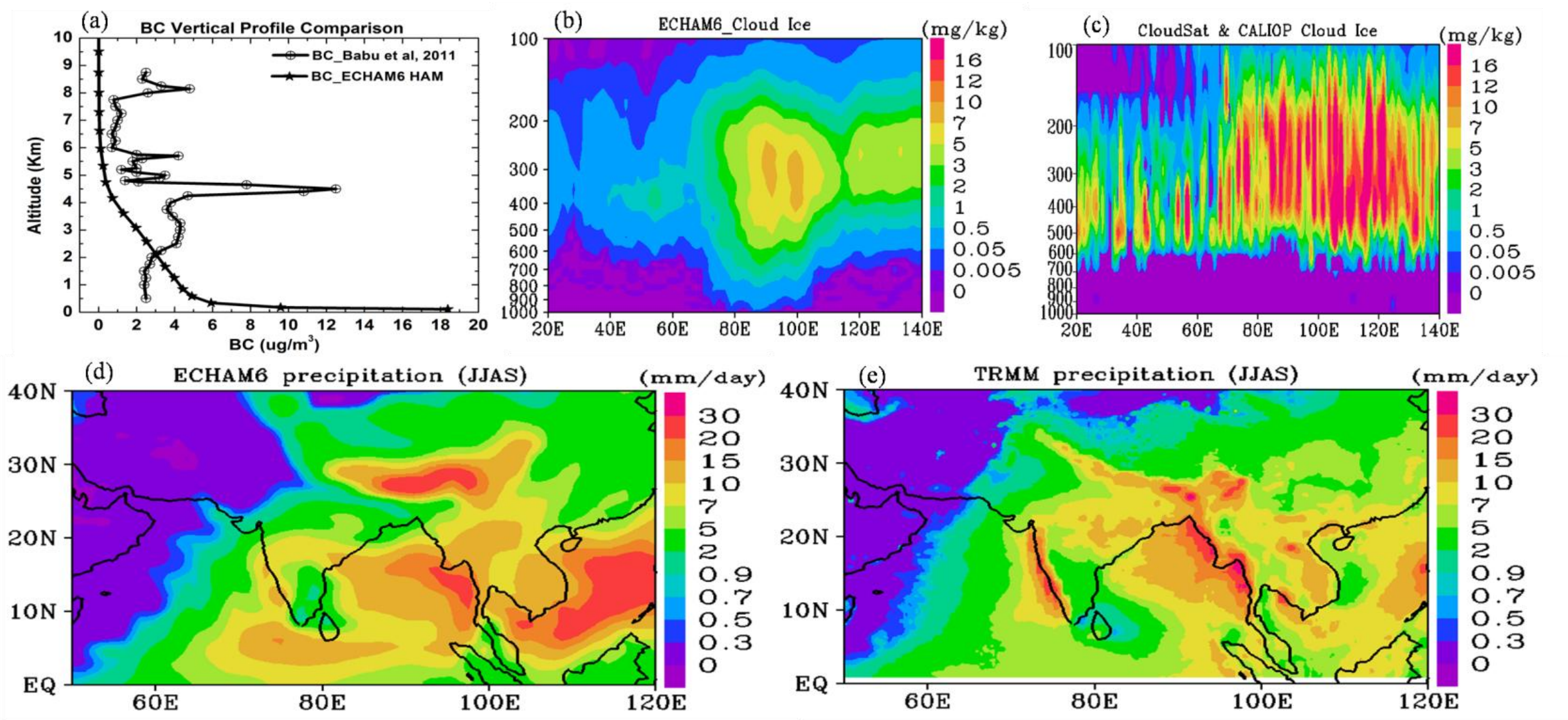

Figure 2: (a) Vertical distribution of BC aerosols (μg/m$^3$) measurements on 17 March 2010 at Hyderabad (17.48 °N,78.40° E) , India (Babu et al., 2011) and ECHAM6-HAM simulated BC aerosols from CTRL simulations avegraed for month of March at a grid centred at 17°N and 78°E, Longitude-pressure distribution of cloud ice mass mixing ratio (mg/kg) averaged for the monsoon season and 20-40°N ( b) ECHAM6-HAM CTRL simulation (c) CloudSat and CALIPSO combined 2C-ICE L3 for the years 2007-2010, (d) seasonal mean precipitation (mm/ day) obtained from (d) ECHAM6-HAM CTRL simulation (e) TRMM averaged for period 1998-2005.

Atmospheric Chemistry and Physics
EGU
Discussions

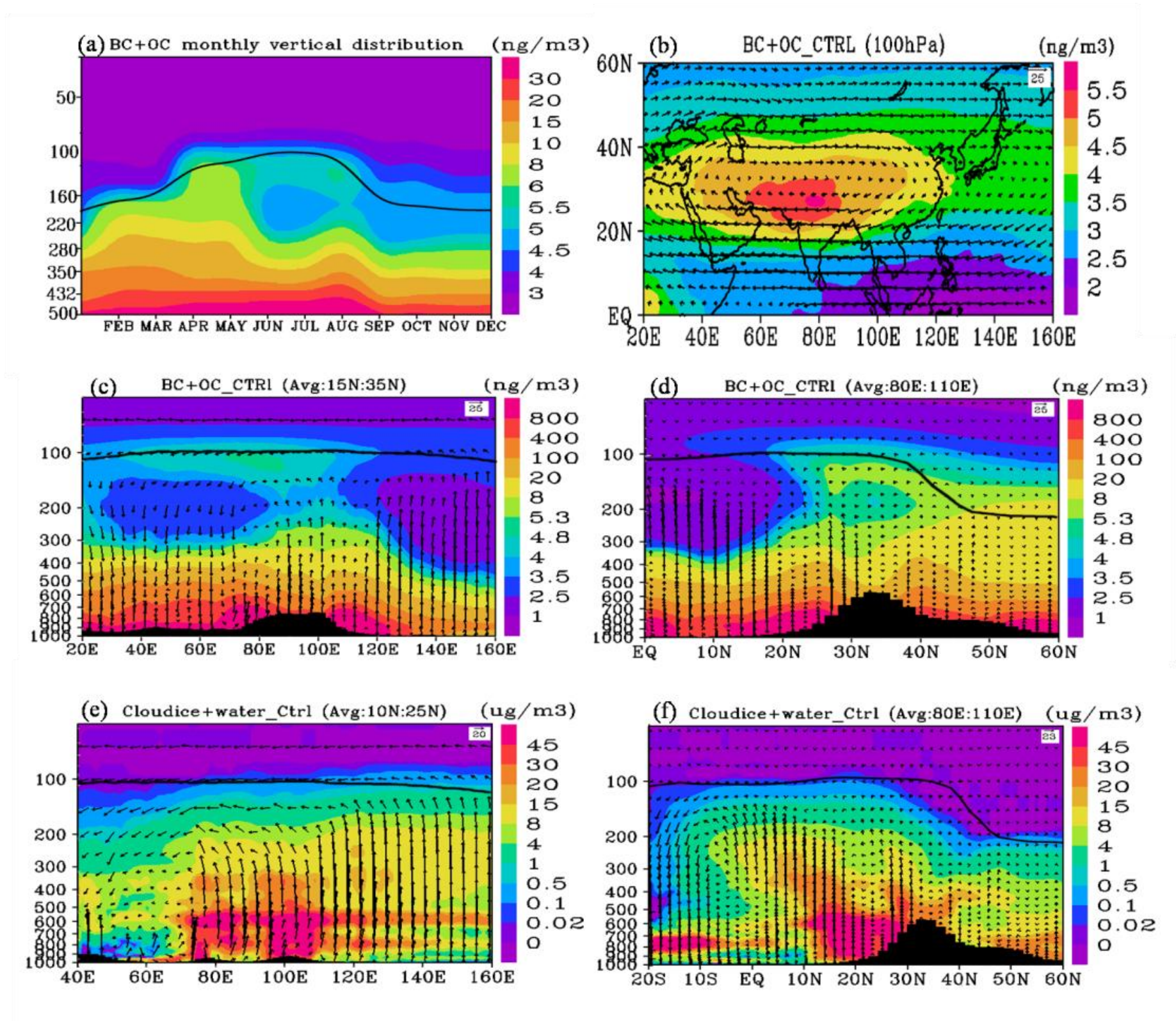

Figure 3: Distribution of BC and OC aerosols (ng/m$^3$) together (a) monthly variations averaged for the region 70°E - 120°E, 25°E - 45°E, (b) averaged for the monsoon season and at 100 hPa, (c) longitude-presure cross section averaged for 15°N - 35°N and monsoon season (d) latitude-pressure cross section averaged for 80°E - 110°E and monsoon season, Distribution of cloud ice+cloud water (µg m$^{-3}$) (e) longitude-presure cross section averaged for 10°N - 25°N and monsoon season (f) latitude-pressure cross section averaged for 80°E - 110°E and monsoon season. Black arrows indicate wind vectors. The vertical velocity field has been scaled by 1000. The black line represents the tropopause. In Figs. (a), (c), (d), (e), (f) tropopause is averaged over the same region where field parameter is averaged.

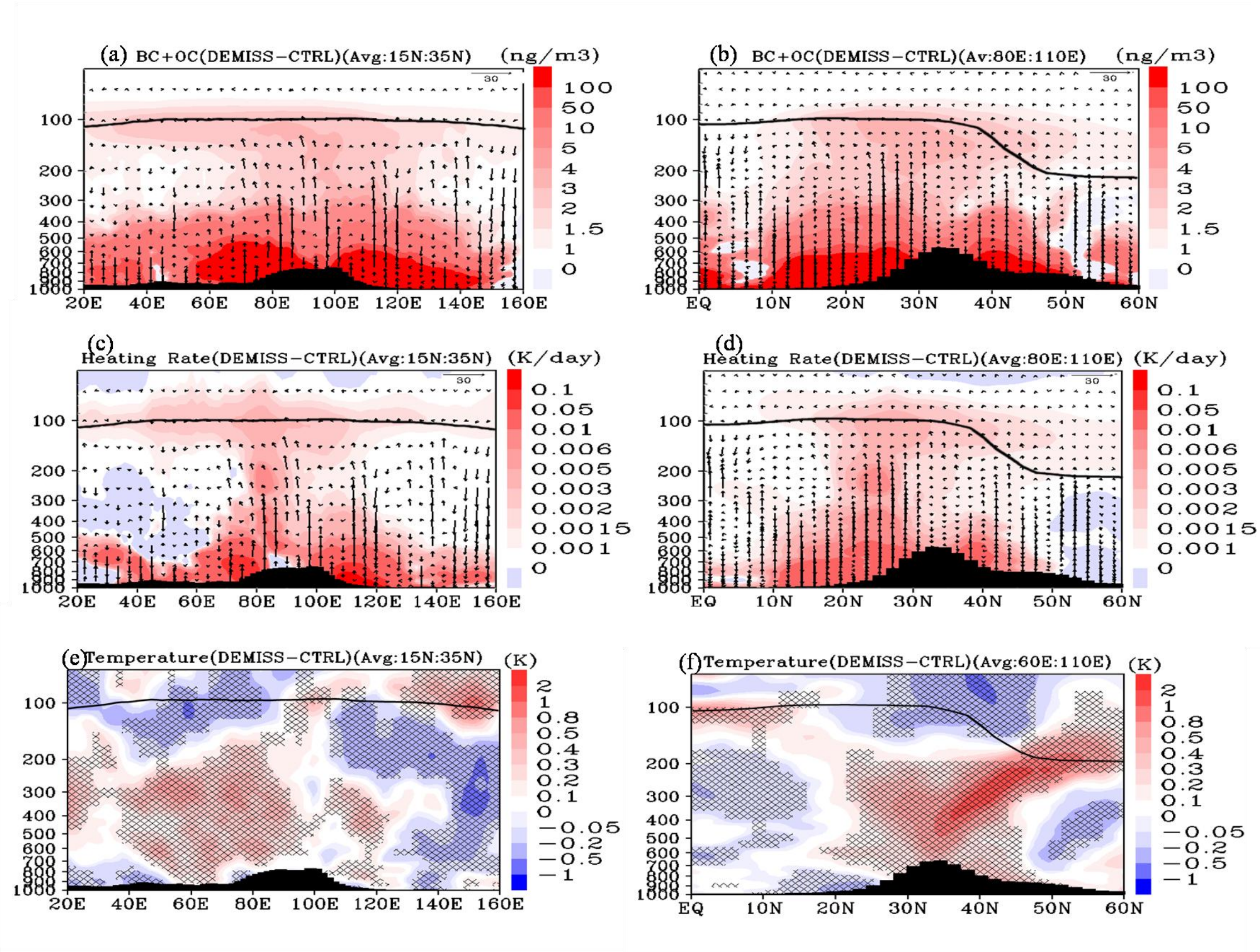

Figure 4: Distribution of anomalies (DEMISS - CRTL), of BC and OC aerosols (ng/m$^3$) together averaged for the monsoon season (a) longitude-pressure cross section (averaged over 15°N - 35°N) (b) latitude-pressure crosssection (averaged over 80ºE-110°E), (c) and (d) same as (a) and (b) but for heating rate anomalies (K/day), Black arrows indicate wind vectors (the vertical velocity field has been scaled by 1000). Distribution of anomalies in temperature (K) (e) longitude-pressure cross section (averaged over 15ºN-35°N), (f) latitude-pressure cross section (averaged over 60°E -110°E). In Figs (e) and (f) black hatched lines indicate 99% confidence level. The black line represents the tropopause. The tropopause is averaged over15°N -35°N for Figs. (a), (c) , (e) and over 80ºE-110°E for Figs. (b), (d) and (f).

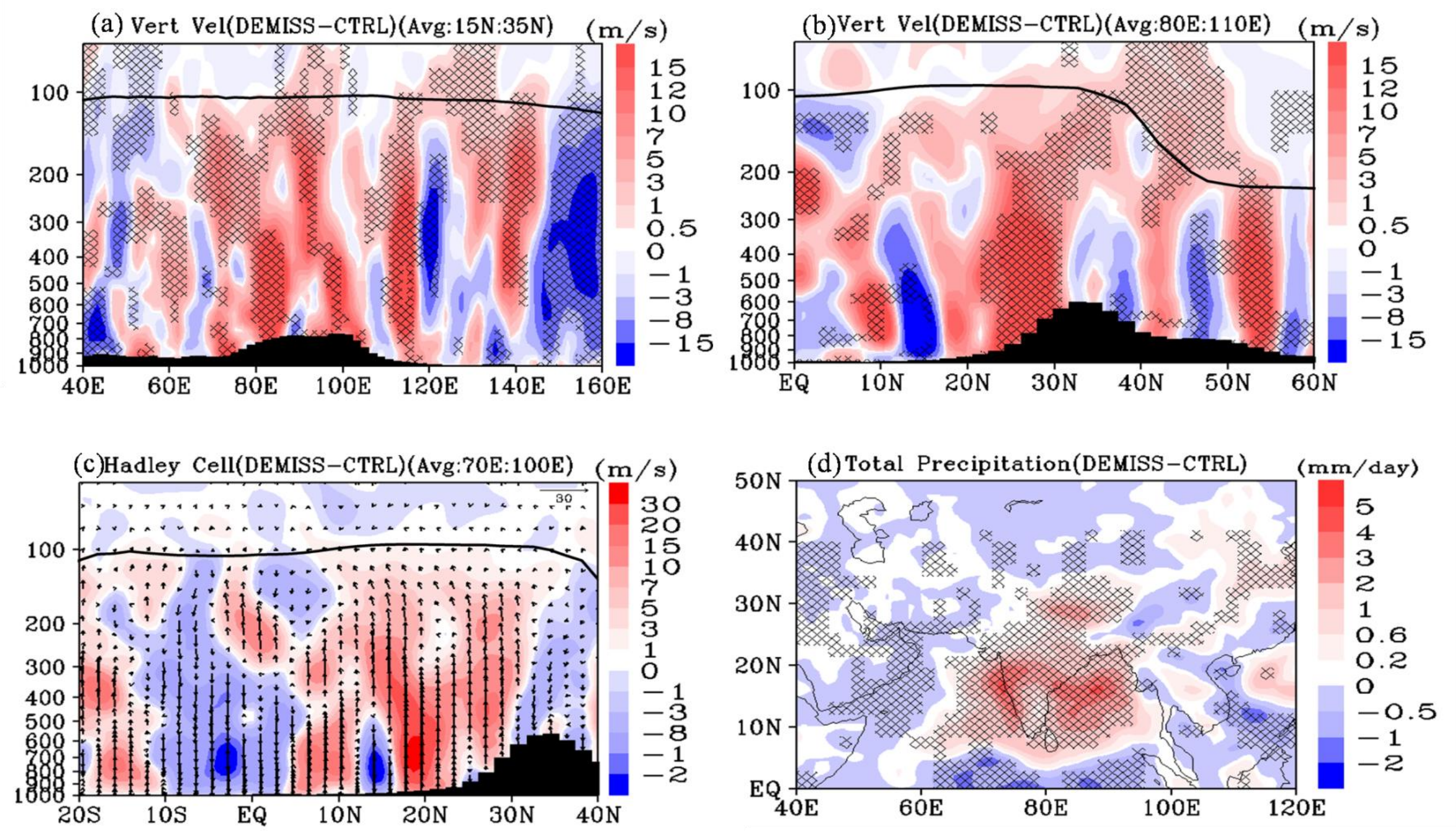

Figure 5: Distribution of anomalies in vertical velocities (m s$^{-1}$) (DEMISS – CRTL) averaged for the monsoon season (a) longitude-pressure (ageraged over 15ºN - 35°N) (b) latitude-pressure distribution of (averaged over 80ºE-110°E), (c) Difference in the meridional circulation due to enhanced carbonaceous aerosols emissions averaged for the monsoon season and over 70ºE-100ºE. Black arrows indicate wind vectors. In Figs (a)-(c) the vertical velocity field has been scaled by 1000 and the thick black line shows the tropopause. The tropopause is averaged over 15°N -35°N for Figs. (a), (c) and over 80ºE-110°E for Fig. (b), (d) distribution of anomalies of total precipitation (mm/day) averaged for the monsoon season. In Figs (a), (b) and (d) hatched lines indicate 99% confidence level.

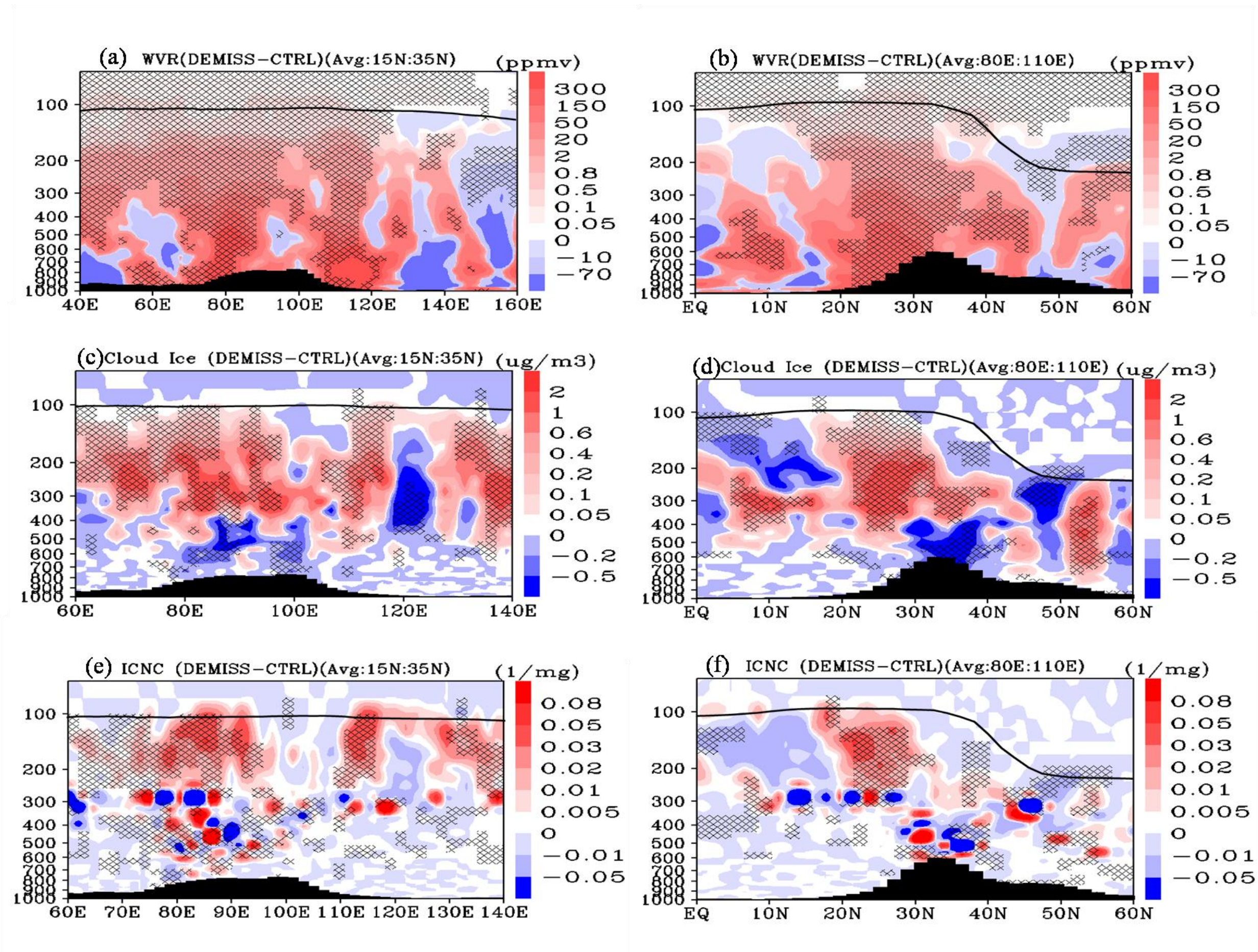

Figure 6: Distribution of anomalies (DEMISS-CTRL) of water vapour (ppmv) averaged for the monsoon season, (a) longitude-pressure cross section (averaged over 15°N - 35°N) (b) latitude-pressure cross section (averged over 80°E - 110°E), (c) and (d) same as (a) and (b) but for cloud ice (μg/m$^3$) and (e) and (f) for ice crystal number concentration (ICNC) (1/mg).The thick black line shows the tropopause while black hatched lines indicate 99% confidence level. The tropopause is averaged over 15°N -35°N for Figs. (a), (c) , (e) and over 80°E-110°E for Figs. (b), (d) and (f).