# Peer review of "Potential impact of carbonaceous aerosol on the Upper Troposphere and Lower 1 Stratosphere (UTLS) and precipitation during Asian summer monsoon in a 2 global model simulation 3 Suvarna Fadnavis1, Gayatry Kalita1, K. Ravi Kumar1, Blaz Gasparini2 and Jui-Lin"

_Atmospheric Chemistry and Physics, 2017_

## Referee Comment (RC1) · Anonymous Referee #1 · 3 Apr 2017

Fadnavis et al. studies the regional impact of carbonaceous aerosol in ASM region by doubling the Asian carbonaceous emissions. In general the topic is interesting and important. However I think the paper may overstate some of the significance and some information are missing/incomplete.

General comments:

1. I am a little bit confused here. The title suggested the paper is going to focus on UTLS region. Reading through the paper, I found little evidence from ECHAM6-HAM model supports a sounding impact of carbonaceous aerosols on UTLS. This study

shows a heating rate of +0.003-0.005K/day (Line 434) due to carbonaceous aerosols near the tropopause, and it is only 1 to 2% of total atmospheric heating rate ($\sim$0.2 to 0.5 K/day). What is the uncertainness of the atmospheric heating rates at the UTLS? Gettelman et al. (2004, Figure 4) suggests that the uncertainties from different radiative transfer models is on the order of +-0.1 k/day, and spatial/temporal distribution of water, ozone, aerosol will add more uncertainties. Is the +0.003-0.005 K/day due to carbonaceous near the UTLS significant enough given the relative large uncertainties.

2. Paper shows one profile BC comparison with balloon sounding. It is hard to tell the concentration from the Figure in linear scale. Maybe a log scale is better for UTLS. In addition, it is necessary to show more model's validations of BC's vertical profile especially in UT since the conclusion relies heavily on modeled BC vertical profile. I know there is a SP2 campaign over Japan up to 9/10 km. You mentioned CARIBIC aerosols, how does your model simulation compared with CARIBIC data?

3. How you treat OC and BC? How much contribution comes from OC and BC separately?

4. Paper shows a warming core above TP, and a large temperature anomaly of 3K (Figure 4f) due to carbonaceous aerosol increase. I wonder is the 3K due to aerosol directly or water (through change in dynamics) or just model noise? The feature (spatial pattern) of 3K temperature anomaly in Figure 4f is different/inconsistent with heating rate/aerosol anomaly in Figure 4(b, d). In addition, your calculated heating rate due to aerosol (Figure 4d) shows some value less than 0.003 K/day, (very difficult to tell from the color scale) above 500 mb, while your 3K feature in Figure 4f is located at regions between 500 and 200 mb. Seems the heating rate (0.003 K/day) is too small to achieve a temperature difference of 3K.

5. Paper mentioned 2.6 W/m2 for the total forcing in Line

Some other suggestions:

a. Figure color scale is hard to tell

b. Line 302, explain why your forcing (+2.7 W/m2) is quite different from other studies from Babu (2002) for +28 W/m2, and Badarinatha and Latha (2006) +42 W/m2. Is that because of different spatial sampling? Please justify.

c. When you show how much vertical velocity or water vapor etc change with your experiment, please also provide % change.

---

## Referee Comment (RC2) · Anonymous Referee #2 · 15 May 2017

This paper considers the impact of doubled BC and OC aerosol emissions in the South Asia monsoon region using the ECHAM6-HAM model. This model has coupling between aerosol and convective processes but lacks important indirect effects and an interactive ocean surface. The authors find a systematic positive impact of doubled BC and OC emissions on the rainfall over India and eastern China. Since it does not account for the sea surface temperature (SST) feedback (e.g. Ganguly et al., 2012) and aerosol indirect impacts, this study should provide additional analysis that helps improve understanding of the impact of aerosols on the atmospheric state.

I recommend this paper for publication after a major revision. In particular, the dynamical response to the aerosol forcing should receive a more thorough analysis and focused discussion. Given the lack of leading order feedback effects in these simulations, aspects that are likely to persist when these feedbacks are included should be given more attention since they have more value. In contrast, the impact on the rainfall is not such an interesting result since it is not likely to be robust given the disagreement with Ganguly et al. (2012).

Major comments:

1) Using fixed SSTs is a major limitation of this model study. The Ganguly et al. (2012) study indicates that surface heating feedbacks have an impact on the rainfall over India. Thus they cannot be ignored. The authors acknowldege this limitation but the value of their study diminishes if atmospheric processes that a exert a leading order impact are neglected. This study would have been of more interest if the process differences between the Ganguly et al. (2012) simulations and the prescribed SST simulations with ECAHM6-HAM had been evaluated.

2) On page 19, lines 405-408, the authors state that "Positive anomalies in cloud ice and ICNC (in the upper troposphere) may be due to enhancement in ASM deep convection...". The model makes it possible to discern such process details in contrast to observational studies. Similarly on page 15, lines 321-323, "may" is used to describe a process that can be diagnosed from the model. The use of "may" is routine in other instances where transport impacts on aerosols and other tracers are considered in this paper. It makes it seem as if the authors are not sure of the transport effects of the doubled BC and OC emissions. I suggest the "may" qualifier be removed and if there is some uncertainty in the interpretation of the model processes, then this uncertainty should be explicitly noted and discussed. Expanding on the aerosol impact on the circulation state would add value to this study given its limitations. For example, is there an organized circulation structure (both diabatic and isentropic) that is characteristic of the South Asia monsoon region? This subject is covered to various degrees in other

studies but lacks the emphasis it deserves in this study. A more focused discussion of the dynamical response to the aerosol forcing is needed and a comparison with the dynamical regime in the Ganguly et al. (2012) study may help to improve understanding of the aerosol sensitivity in this region.

Evaluation of vertical transport in the model is worthwhile in the case of the "sub-grid" convective transport since the convection parameterization contributes heating tendencies that impact the circulation. Figures 3, 4 and 5 have the circulation anomaly vectors presented. In the case of UTLS transport, it is next to impossible to see the transport pattern in many cases since the lower altitude vectors dominate the scaling. Figure 5 compensates for the poor readability of the other figures and justifies the link between enhanced penetration of aerosol into the lower stratosphere and the increased vertical circulation through the tropopause which has substantial statistically significant regions.

3) Figure 5 indicates that it is not just the Hadley circulation which changes but the low altitude jet structure between 10 and 25 N. This has an impact on the rainfall pattern as well.

4) Indirect effects of BC and OC are not included in these simulations. It would have been much more worthwhile to consider the impact of increased IN and the increased cloud evaporation due to aged BC fractions in cloud liquid and ice phases. It is not at all clear that these indirect effects would not substantially change the results. Thus the lack of indirect aerosol forcing is a serious limitation of this study. The authors should include more discussion of this limitation than the cursory mention on page 14. The conclusions of this paper have substantially reduced relevance considering the lack of important feedback processes including SSTs and cloud indirect effects.

Minor comments:

p3. l33: replace "being" by "is", "population and economies" by "regions in terms of population and economy" p3. l35: replace "major" by "main" or "primary" p3. l38:

replace "contribute largely" by "substantially contribute" p3. l41: replace "significantly large" by "large" or "significant" p21. l445: replace "evidences" by "evidence" p21. l446: replace "show" by "shows"

---

## Author Response (AR1)

**Replies to Anonymous Referee #1**

Fadnavis et al. studies the regional impact of carbonaceous aerosol in ASM region by doubling the Asian carbonaceous emissions. In general the topic is interesting and important. However I think the paper may overstate some of the significance and some information are missing/incomplete. General comments:

Reply: We thank reviewer for careful reading and valuable suggestions. We have incorporated all the suggestions given by the reviewer. These changes are marked in red color and corresponding line numbers are indicated.

1. I am a little bit confused here. The title suggested the paper is going to focus on UTLS region. Reading through the paper, I found little evidence from ECHAM6-HAM model supports a sounding impact of carbonaceous aerosols on UTLS.

Reply(1): We have revised the manuscript and incorporated discussion of impact of carbonaceous aerosols on monsoon processes. Therefore the title of the paper is now changedas "Potential impact of Asian carbonaceous aerosol emission on the Upper Troposphere and Lower Stratosphere (UTLS) and precipitation during Asian summer monsoon in a global model simulation".

2. This study shows a heating rate of +0.003-0.005K/day (Line 434) due to carbonaceous aerosols near the tropopause, and it is only 1 to 2% of total atmospheric heating rate (~0.2 to 0.5 K/day). What is the uncertainness of the atmospheric heating rates at the UTLS? Gettelman et al. (2004, Figure 4) suggests that the uncertainties from different radiative transfer models is on the order of +-0.1 k/day, and spatial/temporal distribution of water, ozone, aerosol will add more uncertainties. Is the +0.003-0.005 K/day due to carbonaceous near the UTLS significant enough given the relative large uncertainties.

Reply(2): Heating rate figures are revised. We have now added 95% confidence level and they show that heating rate anomalies are significant in the UTLS (Fig. 4 c-d; page 44).

3. Paper shows one profile BC comparison with balloon sounding. It is hard to tell the concentration from the Figure in linear scale. Maybe a log scale is better for UTLS. In addition, it is necessary to show more model's validations of BC's vertical profile especially in UT since the conclusion relies heavily on modeled BC vertical profile. I know there is a SP2 campaign over Japan up to 9/10 km. You mentioned CARIBIC aerosols, how does your model simulation compared with CARIBIC data?

Reply(3): Thank you for the suggestion. Figure is now modified on log scale. We have tried to obtain the above mentioned data for model's validation. We had contacted Dr. Hang Su, the investigator of CARIBIC and he informed that data is not ready to release (an email correspondence attached). However, additionally, we have compared model simulations with aircraft measurements in the lower-mid troposphere over Guwahati, India since we

could not get BC measurements in the UTLS (Fig. 2a-d and discussion on pages 10-11, lines 204-228).

4. How you treat OC and BC? How much contribution comes from OC and BC separately?

Reply(4): Thank you for the suggestion. We have incorporated a few figures showing contribution of BC and OC separately (Table1; Fig. 7; Fig.S1-Fig.S4) and related discussions are incorporated (Page 15-16, lines 316-321, 325-337, 349-352, page 17, lines 363-369; page 19; lines 419-422).

5. Paper shows a warming core above TP, and a large temperature anomaly of 3K (Figure 4f) due to carbonaceous aerosol increase. I wonder is the 3K due to aerosol directly or water (through change in dynamics) or just model noise? The feature (spatial pattern) of 3K temperature anomaly in Figure 4f is different/inconsistent with heating rate/aerosol anomaly in Figure 4(b, d). In addition, your calculated heating rate due to aerosol (Figure 4d) shows some value less than 0.003 K/day, (very difficult to tell from the color scale) above 500 mb, while your 3K feature in Figure 4f is located at regions between 500 and 200 mb. Seems the heating rate (0.003 K/day) is too small to achieve a temperature difference of 3K. Paper mentioned 2.6 W/m2 for the total forcing in Line Some other suggestions:

Reply(5): Thank you for the suggestion. We have improved the color scale in revised manuscript. These figures show heating rates are ~0.03-0.05 K/day near 500 hPa and temperature ~1K over the Tibetan Plateau (TP). We have mentioned that temperature anomalies of 1K in the upper troposphere over the TP may be due to heating by aerosol and water vapour together. The increase in water vapour in the mid-upper troposphere in response to dynamical changes (as seen in Fig 8a and b) induced by doubling of carbonaceous aerosols contributes additionally to this warming (page 17, lines 363-369 also page 21, lines 459-460).

Figure color scale is hard to tell

a. Reply: We have re-plotted the figures and color scale is improved.

b. Line 302, explain why your forcing (+2.7 W/m2) is quite different from other studies from Babu (2002) for +28 W/m2, and Badarinatha and Latha (2006) +42 W/m2. Is that because of different spatial sampling? Please justify.

Reply: The above stated forcing (+2.7 W/m2) is anomalies obtained from Demiss-CTRL simulations. Studies pertaining to BC/OC radiative forcing are sparse over the Indian region. Radiative forcing given by Babu (2002) for +28 W/m2, and Badarinatha and Latha (2006) +42 W/m2 are at a point location and during different season. This discussion is now moved in the introduction section (page 5, lines 92-97).

We have compared radiative forcing with Sreekanth et al., (2007) during the monsoon season. The reasons for differences are also explained (pages15-16, lines 328-333).

c. When you show how much vertical velocity or water vapor etc change with your experiment, please also provide % change.

Reply: Thank you for the suggestion.The changes in water vapor are provided in % (Fig.8a-b) (related discussion on page 21, lines 451-453). Values of vertical velocity are small and to avoid division by small values, we show differences.

**Replies to Anonymous Referee #2**

This paper considers the impact of doubled BC and OC aerosol emissions in the South Asia monsoon region using the ECHAM6-HAM model. This model has coupling between aerosol and convective processes but lacks important indirect effects and an interactive ocean surface. The authors find a systematic positive impact of doubled BC and OC emissions on the rainfall over India and eastern China. Since it does not account for the sea surface temperature (SST) feedback (e.g. Ganguly et al., 2012) and aerosol indirect impacts, this study should provide additional analysis that helps improve understanding of the impact of aerosols on the atmospheric state. I recommend this paper for publication after a major revision.

Reply: We thank reviewer for careful reading and valuable suggestions. Aerosol indirect impacts were already included in our model simulations. We have clarified it via email communication with model developer (email correspondence attached). There was misunderstanding related to model settings and we are sorry for this confusion. We have added few figures showing indirect impacts of aerosols from model simulations.

We have now incorporated additional analysis to show the impact of aerosols on monsoon processes. The changes are marked in red color and corresponding line numbers indicated.

1) In particular, the dynamical response to the aerosol forcing should receive a more thorough analysis and focused discussion. Given the lack of leading order feedback effects in these simulations, aspects that are likely to persist when these feedbacks are included should be given more attention since they have more value. In contrast, the impact on the rainfall is not such an interesting result since it is not likely to be robust given the disagreement with Ganguly et al. (2012).

Reply(1): Thank you for the suggestion. We have now incorporated discussion on the dynamical response to the aerosol forcing and corresponding figures (Carbonaceous aerosol induced changes in cross-equatorial jet, clouds, static stability from Brunt Vaisala frequency) (Fig. 5a, 5e, Fig. 6; pages 17-18 lines 376-380; page 18-19, lines 395-405 ) in the revised version.

The objectives of current study and Ganguly et al. (2012) are different. Both the studies give important results of impact black carbon aerosols on precipitation. Study by Ganguly et al. (2012) gives impact on precipitation on climate scale (present day and pre-industrial emissions) while the present study gives impact of atmospheric Asian carbonaceous aerosols on precipitation on the seasonal scale. Since purposes of these two studies are different the model set-ups used are different. Ganguly et al. (2012) used a general circulation model coupled to the surface layer of the ocean (slab ocean setup) to understand slow response from SSTs since they want to study impact on climatic scale. In contrast, the current study uses atmospheric-aerosol-climate model. Previous model studies using prescribed SSTs (Chung et al., 2002; Menon et al., 2002; Lau et al., 2006; Randles and Ramaswamy, 2008) also show increase in precipitation over India due to black carbon aerosols. These results are in agreement with present study. This is clarified in the revised manuscript (page 20, lines 427-441).

(2)      Major comments: 1) Using fixed SSTs is a major limitation of this model study. The Ganguly et al. (2012) study indicates that surface heating feedbacks have an impact on the rainfall over India. Thus they cannot be ignored. The authors acknowledge this limitation but the value of their study diminishes if atmospheric processes that a exert a leading order impact are neglected. This study would have been of more interest if the process differences between the Ganguly et al. (2012) simulations and the prescribed SST simulations with ECAHM6-HAM had been evaluated.

Reply(2): As mentioned above objective of the present study and Ganguly et al. (2012) are different and therefore model set-up are different. In the past a number of studies (Chung et al., 2002; Menon et al., 2002; Lau et al., 2006; Randles and Ramaswamy, 2008) has analyzed impact of black carbon from model simulations using fixed SSTs. Their results are consistent with the present study. While in the present study we have studied impact of Asian carbonaceous aerosols unlike global BC emissions documented in the previous studies (page 20, lines 427-441).

3) On page 19, lines 405-408, the authors state that "Positive anomalies in cloud ice and ICNC (in the upper troposphere) may be due to enhancement in ASM deep convection...". The model makes it possible to discern such process details in contrast to observational studies.

Reply(3):  We have removed "may be". This sentence is now changes as "Positive anomalies in cloud ice and ICNC (in the upper troposphere) are due to enhancement in ASM deep convection (increase in heating rates, mid/upper tropospheric temperature, vertical velocity, and monsoon Hadley circulation) induced by the doubling of carbonaceous aerosols emissions" (page 22, lines 471).

4) Similarly on page 15, lines 321-323, "may" is used to describe a process that can be diagnosed from the model. The use of "may" is routine in other instances where transport

impacts on aerosols and other tracers are considered in this paper. It makes it seem as if the authors are not sure of the transport effects of the doubled BC and OC emissions. I suggest the "may" qualifier be removed and if there is some uncertainty in the interpretation of the model processes, then this uncertainty should be explicitly noted and discussed.

Reply(4): As suggested "may" is removed from the discussions in the revised manuscript.

5) Expanding on the aerosol impact on the circulation state would add value to this study given its limitations. For example, is there an organized circulation structure (both diabatic and isentropic) that is characteristic of the South Asia monsoon region? This subject is covered to various degrees in other studies but lacks the emphasis it deserves in this study. A more focused discussion of the dynamical response to the aerosol forcing is needed and a comparison with the dynamical regime in the Ganguly et al. (2012) study may help to improve understanding of the aerosol sensitivity in this region.

Reply(5): As mentioned in reply (1), we have now incorporated discussion on the dynamical response to the aerosol forcing and corresponding figures (carbonaceous aerosol induced changes in cross equatorial jet, clouds, static stability) (Fig. 5a, 5e, Fig. 6; pages 17-18 lines 373-377; page 18-19, lines 392-402 ) in the revised version since changes in circulation and atmospheric heating and temperature are already presented in the previous version.

As mentioned in reply (1). The purpose of the present study is different than Ganguly et al (2012) and therefore model set-ups are different. We have included discussions on this (page 20, lines 427-442).

6) Evaluation of vertical transport in the model is worthwhile in the case of the "subgrid" convective transport since the convection parameterization contributes heating tendencies that impact the circulation. Figures 3, 4 and 5 have the circulation anomaly vectors presented. In the case of UTLS transport, it is next to impossible to see the transport pattern in many cases since the lower altitude vectors dominate the scaling. Figure 5 compensates for the poor readability of the other figures and justifies the link between enhanced penetration of aerosol into the lower stratosphere and the increased vertical circulation through the tropopause which has substantial statistically significant regions.

Reply (6): Figure 5 is now improved to show transport pattern.

7) Figure 5 indicates that it is not just the Hadley circulation which changes but the low altitude jet structure between 10 and 25 N. This has an impact on the rainfall pattern as well.

Reply(7): As suggested, discussion on the low level jet structure between 10°N and 25° N is now incorporated (page 18, lines 395-397).

8) Indirect effects of BC and OC are not included in these simulations. It would have been much more worthwhile to consider the impact of increased IN and the increased cloud

evaporation due to aged BC fractions in cloud liquid and ice phases. It is not at all clear that these indirect effects would not substantially change the results. Thus the lack of indirect aerosol forcing is a serious limitation of this study. The authors should include more discussion of this limitation than the cursory mention on page. The conclusions of this paper have substantially reduced relevance considering the lack of important feedback processes including SSTs and cloud indirect effects.

Reply(8): Aerosol indirect impacts were already included in our model simulations. We have clarified it via email communication with model developer (email correspondence attached). There was misunderstanding related to model settings and we are sorry for this confusion. We have added a figure (Fig 6) showing indirect impacts of aerosols from model simulations. Section 5 'Conclusion' section is now changed as 'Summary and conclusions'

 Minor comments:

p3. l33: replace "being" by "is", "population and economies" by "regions in terms of population and economy"

Reply: The above suggestion is incorporated in the revise the manuscript at page 3 line 35-36

p3. l35: replace "major" by "main" or "primary"

Reply: The above suggestion is incorporated in the revise the manuscript at page 3 line 38

p3. l38: C3 ACPD Interactive comment Printer-friendly version Discussion paper replace "contribute largely" by "substantially contribute"

Reply: The above suggestion is incorporated in the revise the manuscript at page 3 line 39-40

p3. l41: replace "significantly large" by "large" or "significant"

Reply: The above suggestion is incorporated in the revise the manuscript at page 3 line 43

p21. l445: replace "evidences" by "evidence"

Reply: The above suggestion is incorporated in the revise the manuscript at page 24 line 526

p21. l446: replace "show" by "shows"

Reply: The above suggestion is incorporated in the revise the manuscript at page 24 line 527

---

## Author Response (AR2)

**Replies to review's Comments**

Review of Paper: Potential impact of carbonaceous aerosols on the Upper Troposphere and Lower Stratosphere (UTLS) and precipitation during Asian summer monsoon in a global model simulation by Fadnavis et al. 2017, submitted to Atmospheric Chemistry and Physics

In general the paper is improved. However there are some issues that needed to be addressed before publication, especially Major concern #1.

Reply: We thank reviewer for valuable suggestions and careful reading. All the suggestions given by the reviewer are incorporated in the revised version. These changes are indicated at line numbers mentioned below and also marked in red color.

Major concerns:

 In the previous paper version, the increased heating rate due to BC/OC near the tropopause in the ASM region was 0.003-0.005 K/day; however in the revised the version, the heating rate is 0.02-0.03 K/day, which is a factor of 6 larger. The heating rate of CTRL run remains the same between two versions of the paper. The BC/OC amount in UTLS remains similar (about 2 ng/m3). I wonder what happened? What have you changed? Also I noticed your TOA radiative forcing changed from strong negative to positive...Please explain.

Reply (1): In the previous version heating rates were estimated from model runs of a few years. Since the parameters, 'heating rates' and 'radiative forcing' were not extracted in output stream for every year to save space on the system with the thought that there may be minor changes in their values. We thank the reviewer for pointing out uncertainty in heating rate during last revision. To show significance level we have extracted these parameters for all the years which have improved the values and they are significant at 99% confidence level. Since Tibetan Plateau (TP) and Indo-Gangetic Plains (IGP) play important role in driving monsoon Hadley circulations we show radiative forcing over this region while in the first version of the manuscript it was over the ASM region.

We have seen that there is year to year variability in BC and OC concentration at different levels over the region of interest TP and IGP. This may be related to transport pattern due to monsoon convection in response to varying SST. These yearly varying concentrations of BC/OC aerosols will change RF and heating rates. Therefore RF and heating rates estimated from 20 years of simulations will show improved values. These values show better agreement with other studies (Myhre et al., 2013; Sreekanth et al., 2007). To show variability in RF we have incorporated stranded deviation values in table-1.

 The authors added more model experiments: add "double OC only run" and double "BC only run". Line 366-369: the radiative forcing for each case do not add up (compared with "double OC and BC run"). I can imagine the system is non-linear, but please provide more discussions on what cause the non-linearity. Reply(2): Thank you for the suggestion. We have incorporated discussions on non-linearity in the radiative forcing due to BC/OC (pages 16-17, line nos 354-356).

Line 363-369: I am not convinced by the discussion on the temperature anomaly core in figure 4f. It is good to see in FigureS3a, the temp anomaly core is in the "double BC only run". I am still wondering which leads to the warming core, note the core in the figure extends to mid-high latitudes. The discussions in the paper are not enough, and seems the authors are somehow guessing. Please justify.

Reply(3): BC aerosols play a major role in creating a ware core. An Extension of warm core to mid-high latitudes is due to BC emission from mid latitudes regions of China, Mongolia, Russia within to 80-110E. We have now incorporated discussion on this (page 17, line nos 375-377).

(4) Line 348: ASM region, not ASM

Reply(4): It is now corrected in the revised manuscript (page 16, line no 349).

(5) Line 216-217: re-write the sentence. I assume you meant the values range from surface to 10 km in model, while 6 km in observations.

Reply(5) : It is re-written in the revised manuscript (page 10-11, line nos 216-218).

(6)Line 366: Figure 8 instead of Figure 7?

Reply(6): Yes, it is figure 8. It is now corrected in the revised manuscript (page 17, line no 372).

(7) Please clarify the confidence level used for multiple figures. Are they 99% or 95%? In the response, you mentioned 95% while 99% in the figure captions.

Reply(7): In the revised version significance is plotted are 99% confidence level.

---

## Author Response (AR3)

**Replies to Co-Editor's comments**

We thank Co-Editor for careful reading and suggestions. We have incorporated all the suggestions given by the Co-Editor. Changes are marked in red colour and corresponding line numbers are indicated below.

(1). I am still not convinced by the discussion on the temperature anomalycore in figure 4f.

The authors do not show sufficient discussion. Simple sentence in the paper"This warm core extends to mid-high latitudes. It is related to BC emissionfrom regions of China, Mongolia, Russia" is not sufficient. I don't see anyaerosol enhancement that is consistent with this feature in mid-highlatitude. Please investigate more on the cause, how black carbon leads to the warming pattern and how robust it is.

Reply(1): Thank you for the suggestion. For our sensitivity simulations, we have increased both BC and OC over the South East Asian region  $(10^{\circ}S - 50^{\circ}N; 65^{\circ}E - 155^{\circ}E)$ . This region includes China, Mongolia, and Russia etc. There is transport from these countries to the upper troposphere. We have provided a supplementary figure (Fig. S4) showing transport of black carbon, from regions of China, Mongolia, and southern Russia, from surface to 200 hPa, extending to mid-latitudes.

To show warming over the Tibetan plateau due to enhanced carbonaceous aerosols, we have shown latitude-pressure cross section ( $80^{\circ}E -110^{\circ}E$ ) in figure 4f. This figure also shows wariming over 40-48°N from surface to ~200 hPa. This warming is related to upward and northward transport of BC from China, Mongolia, and southern Russia to mid latitudes as shown in supplementary figure (Fig. S4). Figure 8b shows negative water vapour anomalies in the same region. In the UTLS water vapour causes cooling and thus a decreased radiative cooling, this might have partially contributed towards the warming. It is now mentioned in the revised manuscript (Page 17-18, line Nos.378-383).

We have tried to plot BC distribution from CTRL simulations and BC anomalies from DBConly-CTRL simulations (Figure 1a-b shown below). Figure 1a shows upward and northward transport from regions of China, Mongolia, and southern Russia to the mid-latitude upper troposphere. However, this transport is not very clear near 350-200 hPa in DBConly-CTRL since differences are not linear along the path way. This may be due to non-linear heating due to doubling of BC. Hence we have shown figure 1a as supplementary figure S4.

Figure 1: Latitude pressure cross section averaged for  $(80^{\circ}\text{E} - 110^{\circ}\text{E})$  and for the monsoon season for (a) BC aerosols (ng m-3) from CTRL simulations and (b) anomalies of BC aerosols (ng m-3) from DBConly-CTRL simulations.

(2). Abstract: "This increases precipitation amounts over India and northeast China." By how much? Confidence level?

Reply(2): Thank you for the suggestion. The quantitative estimates are now provided in the abstract and confidence level is also mentioned (Page 2, line Nos.29-30).

(3). Abstract: "Doubling of emissions of BC and OC aerosols, each, over theSouth East Asia  $(10^{\circ}\text{S} - 50^{\circ}\text{N}; 65^{\circ}\text{E} - 155^{\circ}\text{E})$  show that lofted aerosolsenhance radiative heating rates (0.02-0.03 K/day) near the tropopause,produce significant warming (1K), and instability in the mid/uppertroposphere." Warming of 1K? where (lat, lon, tropopause?)? I see 0.5-1Kwarming 0-10N tropopause, is it what you claimed here? How robust is it?

Reply(3): As suggested above sentence is re-written (Page 1, line No. 20). We have shown 99% confidence levels in most the figures (Figs. 4 - 8). This indicates that the results are robust.

---

## Author Response (AR4)

Reply to the Editor's comment:

(1) Thanks very much for your revisions of the paper. I think you have addressed the remaining points, but there is one question from my side that has arisen in your reply and revisions. You say that increased water vapour in the UTLS should lead to surface cooling (if I understand you correctly). I think this effect is correct in principle but is not applicable at UTLS altitudes but rather at say 10 hPa. At UTLS altitudes increased water vapour should lead to surface warming (see e.g. Fig. 1 in Riese et al., JGR, 2012). Perhaps you could clarify my confusion at this point.

Reply: We thank the Editor for the comment. In the manuscript we have not linked the increased water vapour in the UTLS to the surface cooling. The reference (Riese et al., JGR, 2012) provided by the Editor states that increase in water vapour in the UTLS increases the surface temperature. However, in the present manuscript we explained warming over the Tibetan Plateau and mid latitude at the altitudes between 300-200 hPa in response to enhanced carbonaceous aerosols transported in the UTLS. The warming in the UTLS (200-300 hPa) is partially due to transport of enhanced black carbon (BC) aerosols and water vapour at those pressure levels.

We have stated that "It is related to warming by BC aerosols and partially by water vapour. Figure S4 shows that BC aerosols are transported upward and northward from (latitudes 40°N - 48°N) regions of China, Mongolia, and southern Russia to the mid latitude upper troposphere which may contribute to this warming. In addition, the negative water vapour anomalies in the same region (in the UTLS of the mid latitude) seen in Figure 8b imply a decreased radiative cooling, which might have partially contributed towards the warming anomaly."

To show radiative cooling by the water vapour in the UTLS, we have plotted 'net radiative heating rates due to water vapour' from the model (figure-1 below). It shows negative heating rates (indicating cooling) due to water vapour in the upper troposphere and stratosphere. Also numbers of references below (Foster and shine, 1997; Gierens and Eleftheratos, 2016; Clough and Iacono, 1995) also support the model results (cooling of the upper troposphere and stratosphere by the water vapour present there). The figure-1 and references (below) justify our statement from the previous manuscript "negative water vapour anomalies in the same region (in the UTLS of the mid latitude) seen in Figure 8b imply a decreased radiative cooling, which might have partially contributed towards the warming anomaly (Pages 17-18, Line Nos. 378-384).

Figure 1: Latitude-pressure cross section at (95°E) of net (shortwave plus longwave) water vapour heating rate averaged for the monsoon season (JJA) from a reference ECHAM6-HAM2 model's simulation.

**References**

- (1) Foster and shine (1997): (section 3.5)→ The effect of an increase in lower stratospheric water vapor has also been examined, using the 0.5 to 1% per year stratospheric water vapor increases between 1981-1994 observed over Boulder, Colorado [Oltmans and Homann,1995]. The water vapor increases (Figure 13) cause stratospheric cooling which are largest near the tropopause. Coolings of 0.3 K per decade are found at about 18 km.
- (2) Klaus Gierens and Kostas Eleftheratos, Upper tropospheric humidity changes under constant relative humidity, Atmos. Chem. Phys., 16, 4159–4169, 2016 →
  Although the amount of water vapour in these layers (upper troposphere and stratosphere) is only a small fraction of its total amount in the atmosphere, the contribution of water vapour in the upper troposphere to radiative cooling of the atmosphere (locally) is disproportionately large (Clough et al., 1992).
- (3) Clough S. A. and Iacono M. J., Line-by-line calculation of atmospheric fluxes and cooling rates 2. Application to carbon dioxide, ozone, methane, nitrous oxide and the halocarbons JOURNAL OF GEOPHYSICAL RESEARCH, VOL. 100, NO. D8, PAGES 16,519-16,535, AUGUST 20, 1995 →

The water vapor pure rotation region remains of critical importance with respect to the outgoing longwave radiation and to the cooling in the middle and upper troposphere.